# Timing and Modulation of Activity in the Lower Limb Muscles During Indoor Rowing: What Are the Key Muscles to Target in FES-Rowing Protocols?

**DOI:** 10.3390/s20061666

**Published:** 2020-03-17

**Authors:** Taian M. Vieira, Giacinto Luigi Cerone, Costanza Stocchi, Morgana Lalli, Brian Andrews, Marco Gazzoni

**Affiliations:** 1Laboratorio di Ingegneria del Sistema Neuromuscolare (LISiN), Dipartimento di Elettronica e Telecomunicazioni, Politecnico di Torino, 10129 Torino, Italy; giacintoluigi.cerone@polito.it (G.L.C.); marco.gazzoni@polito.it (M.G.); 2PolitoBIOMed Lab, Politecnico di Torino, 10129 Torino, Italy; 3School of Engineering, University of Warwick, Coventry CV4 7AL, UK; brian.andrews@nds.ox.ac.uk; 4Nuffield Department of Surgical Sciences, University of Oxford, Oxford OX1 2JD, UK

**Keywords:** rowing, electromyography, muscle, functional electrical stimulation

## Abstract

The transcutaneous stimulation of lower limb muscles during indoor rowing (FES Rowing) has led to a new sport and recreation and significantly increased health benefits in paraplegia. Stimulation is often delivered to quadriceps and hamstrings; this muscle selection seems based on intuition and not biomechanics and is likely suboptimal. Here, we sample surface EMGs from 20 elite rowers to assess which, when, and how muscles are activated during indoor rowing. From EMG amplitude we specifically quantified the onset of activation and silencing, the duration of activity and how similarly soleus, gastrocnemius medialis, tibialis anterior, rectus femoris, vastus lateralis and medialis, semitendinosus, and biceps femoris muscles were activated between limbs. Current results revealed that the eight muscles tested were recruited during rowing, at different instants and for different durations. Rectus and biceps femoris were respectively active for the longest and briefest periods. Tibialis anterior was the only muscle recruited within the recovery phase. No side differences in the timing of muscle activity were observed. Regression analysis further revealed similar, bilateral modulation of activity. The relevance of these results in determining which muscles to target during FES Rowing is discussed. Here, we suggest a new strategy based on the stimulation of vasti and soleus during drive and of tibialis anterior during recovery.

## 1. Introduction

With the aid of functional electrical stimulation (FES), people with spinal cord injury can exercise at intensities greater than those associated with upper body exercising alone [1,2]. The greater metabolic demand resulting from the combination of electrically and voluntarily elicited contractions likely motivated the emergence of different FES protocols, in particular, the FES-indoor Rowing (henceforth FES Rowing). Notwithstanding the documented musculo-skeletal and cardiovascular benefits of FES Rowing in paraplegia (e.g., decreased loss of bone mineral density, increased aerobic capacity; [3,4,5,6]), no consensus seems to exist as per which muscles to stimulate and when. In FES-Rowing studies, the cyclic flexion-extension of the knee has been respectively achieved through stimulation of quadriceps and hamstrings [1,3,7,8]; often though, the specific quadriceps and hamstrings muscles considered for stimulation are not reported. The stimulation of these two muscle groups was likely intuitively motivated by their mechanical action, necessity of design, and the desire for simplicity in the number of stimulation channels and surface electrodes. This, however, may not be the appropriate set of muscles to target during FES Rowing, as evidenced by: (i) The need of using springs or bungee cord or even by inclining the track to assist recovery, given the difficulty of getting a pure knee flexion [2,5]; (ii) the rapid instalment of fatigue, limiting exercise duration; (iii) the need to fix the trunk, which is the most obvious discrepancy between FES and normal rowing, leading to a short stroke length [8].

Previous research on muscle activity during rowing has not specifically addressed the timing and activity of a large muscle set in relation to the rowing cycle and body side. On one hand, surface electromyograms (EMGs) collected from several muscles consistently revealed the existence of three muscle synergies during indoor rowing [9,10,11]. On the other hand, although all muscles studied were observed to be elicited during rowing, specific information on the timing of muscles’ recruitment and on whether recruitment takes place similarly bilaterally were not reported [9,10,11,12,13]. Moreover, studies focused on the timing of muscle activity during indoor rowing were either limited to a small set of muscles, often not in the lower limbs, did not assess muscles bilaterally, or assessed activation during all-out 2000 m efforts [14,15,16,17]. Instants of muscle activation quantified unilaterally or during intense rowing efforts may not translate appropriately to FES-Rowing applications, given that asymmetries in foot force were reported during indoor rowing [18,19] and that muscle activity seems to change with rowing intensity [11,20]. Identifying when and which leg muscles contribute symmetrically to the rowing cycle would therefore, prove valuable both for manual, automatic, and semi-automatic FES-Rowing protocols, particularly for the FES-Rowing technique in which the trunk is allowed to rotate in the sagittal plane to produce more normal rowing motion [21].

In this stage we do not explicitly describe new sensors or signal processing techniques but investigate which, how many, when, and how different leg muscles are recruited during indoor rowing, on which the future design of sensors, signal processing, and FES control of rowing may be based. From surface EMGs collected bilaterally from eight leg muscles, we specifically ask whether EMG amplitude increases over the background, noise level within the rowing cycle. We characterise instances of activation by quantifying the onset of muscle activation and deactivation and the duration of muscle activity. Given the evidence on side differences in the degree of activity of back muscles during indoor rowing [12,17], we further assess whether the timing and modulation of EMG amplitude differs between legs. At this stage in our research, we exclusively assess the leg muscles because, at the moment, the trunk is constrained during FES-Rowing. From current results we provide specific indication on which muscles to target in FES-Rowing protocols and when to stimulate these muscles, hoping this information may help optimise the health benefits [3,5,6] and maximise the performance [8] of FES-Rowing.

## 2. Material and Methods

### 2.1. Participants

Twenty rowers, two females, were recruited to participate in this study after providing written consent (range: 18–30 years; 170–190 cm; 60–92 kg). All participants have been engaged into national competitions for at least four years. Two of them participated in international competitions; fourth place and sixth place in the coxless four and in the eight, respectively. None of the subjects presented musculo-skeletal injuries and none reported discomfort (e.g., back pain) at the occasion of experiments. Experimental procedures conformed with the *Declaration of Helsinki* and were approved by the Local Ethics Committee (Prot. N. 00/0610; ASL 1—Torino, Italy).

### 2.2. Experimental Protocol

Before commencing experiments, subjects were asked to warm up for roughly 5 min on an indoor rowing machine (Model E, Concept II, Morrisville, NC, USA). Participants could change freely the drag factor of the rowing machine. After warm up, three trials were applied: Subjects were asked to row for 120, 100, and 80 s at 18, 24, and 28 strokes/min (spm). These durations ensured subjects performed a total of at least 30 consecutive strokes per trial and these stroke rate values were selected to cover the range of rowing speed often reported in FES-Rowing training [7,8,22]. Trials were applied at random order with 5 min break in-between. During experiments, the drag factor control on the air damper was not allowed to change, and participants were asked to row as if they were engaged in a regular training session. Visual feedback on stroke rate and instantaneous speed was provided on the rowing machine PM5 display; they were instructed to keep consistently low the time taken to cover 500 m throughout each of the three trials.

### 2.3. EMG and Rowing Machine Recordings

Surface EMGs were collected with circular, bipolar electrodes (15 mm diameter; CDE electrodes, OTBioelettronica, Turin, Italy). After shaving and cleaning the skin with abrasive paste (EVERI; Spes Medica, Battipaglia, Italy), a total of 16 couples of self-adhesive electrodes were positioned, each on each of the following muscles: Tibialis anterior, gastrocnemius medialis, soleus, vastus lateralis and medialis, rectus femoris, biceps femoris, and semitendinosus of both limbs. The centre of each couple of electrodes was positioned at specific muscle locations, in agreement with the SENIAM recommendations [23]. Electrodes were aligned in the proximo-distal direction rather than parallel to the muscle fibres. Given the pennate architecture of the muscles we assessed, the proximo-distal alignment ensured electrodes sampled from a presumably greater number of fibres than if they had been aligned along fibres in a specific proximo-distal location [24,25,26]. For this same reason, to ensure a greatest number of motor units would be represented in the bipolar EMGs collected from each muscle, the centre-to-centre distance between electrodes was 35 mm; this distance has been shown to provide a more representative EMG for the whole gastrocnemius muscle with negligible crosstalk from soleus [27].

Bipolar EMGs were amplified by 180 V/V (10-500 Hz antialiasing filter bandwidth), digitised at 2048 Hz (16 bit A/D resolution) and transmitted via Bluetooth to a computer running on Windows operating system (DuePro EMG and biomechanical systems; LISiN and OTBioelettronica, Turin, Italy). The position of the handle of the rowing machine was measured with an incremental, optical rotary encoder (1024 pulses per revolution, WDG 24C, Wachendor Automation, Geisenheim, Germany). The encoder was mounted coaxially to the flywheel rotation axis [28], providing a resolution of 0.22 mm. Position data was sampled synchronously with the surface EMGs.

### 2.4. Estimating the Timing and Side-Differences in the Modulation of Muscle Activity

Raw surface EMGs were initially visually inspected to identify and discard channels presenting electrode contact issues or large movement artefacts. Surface EMGs were then bandpass filtered with a digital second order Butterworth digital filter (20–400 Hz 6 dB cutoff) in the forward and reverse time direction to avoid phase distortion, full-wave rectified, and then lowpass digital filtered at 5 Hz with a similar filter (Figure 1). These smoothed EMGs, i.e., EMG envelopes, were considered to identify periods of muscle activity. First, we defined the baseline level as five times the standard deviation of the envelope of EMGs collected while subjects were at rest [29]; 5 s of raw EMGs were collected with the subjects sitting on the rowing machine as relaxed as possible while the examiner ensured action potentials of the single motor unit could be observed for the 16 muscles assessed. The onset of muscle activation and silencing was computed by linearly interpolating EMG envelopes when, after the finish instant, these envelopes respectively, first and last equalled the baseline (cf. squares and circles in Figure 1). This procedure provided several onset values per muscle, corresponding to the number of rowing cycles performed per trial. The duration of activity corresponded to the algebraic sum of consecutive intervals between silencing and activation whereas the amplitude of the envelope within these periods was considered to access left and right side-differences in the modulation of muscle activity.

Timing and modulation of muscle activity were assessed in relation to the rowing cycle. Individual cycles were identified from the position of the handle of the rowing machine; catch and finish corresponded respectively to instants of minima and maxima of handle position [28]. Handle position and EMG envelopes were segmented into individual cycles and interpolated to provide an equal number (*N* = 100) of samples per cycle. Onsets and duration of activity were both normalised with respect to the corresponding cycle duration and averaged. Similarly, EMG envelopes within periods of activity were averaged across cycles, providing a single EMG envelope per muscle, from when it was recruited to when it was silenced. Figure 2 shows an example of averaged handle position data and EMG envelopes for each muscle and body side.

### 2.5. Statistics

The given data distribution was Gaussian (Shapiro-Wilk test, *P* > 0.08 in all cases) and the homogeneity of variance (Levene’s test; W > 0.2 in all cases), parametric statistics were applied to assess differences in onset and duration values. A three-way ANOVA arrangement was applied, with body side as repeated measures (three stroke rates × eight muscles × two body sides; [30]). Whenever a main effect was observed, pairwise comparisons were assessed with Bonferroni correction. Similarity in the left-right modulation of activity for each muscle was assessed through the Pearson correlation coefficient and the slope of regression lines; a significant correlation indicates the amplitude of left-right envelopes changes synchronously whereas regression lines with unit slopes indicate the amplitude changed by equal amounts. Before computing Pearson correlation, left and right envelopes were interpolated (N = 100 data points), from recruitment to silencing, and normalised with respect to the maximal value across all cycles for each leg and subject to suppress the effect of spurious factors (e.g., side-differences in electrode-skin impedance; cf. [31]).

## 3. Results

All participants rowed at the requested cadences. The mean (± standard deviation) duration of the rowing cycle across participants was 3.23 ± 0.20, 2.57 ± 0.20, and 2.20 ± 0.09 s for 18, 24, and 28 spm, respectively. The standard deviation across rowing cycles varied from 0.3 to 5.6 mm for handle position (averaged over 100 data points within cycles; *N* = 60; 20 subjects × three stroke rates) and from 2.7% to 6.3% (w.r.t. the duration of rowing cycle) for the onset values (N = 960; 20 subjects × three stroke rates × eight muscles × two body sides). Given no effect of stroke rate on onset values was observed (ANOVA main effect, *P* > 0.05), onset values were averaged across the three stroke rates.

### 3.1. Timing and Duration of Muscle Activity During Rowing

Different muscles were recruited and silenced at different instants during rowing (Figure 3; ANOVA main effect for muscle; *P* < 0.005; *N* = 160, 20 subjects × eight muscles). While quadriceps, gastrocnemius, and soleus were recruited just before catch, the other muscles were recruited both sooner (semitendinosus and tibialis anterior) and later (biceps femoris; Bonferroni post-hoc tests, *P* < 0.02 in all cases). While only tibialis anterior was silenced prior to catch, vastus lateralis and medialis were silenced just before finish and significantly after semitendinosus, biceps femoris, gastrocnemius and soleus (*P* < 0.005). Rectus femoris was silenced later than any other muscle, after catch (*P* < 0.005). These results apply equally to both legs (main effect for body side: *P >* 0.1; *N* = 40, 20 subjects × two sides). Similarly, the duration of activity within rowing cycles was muscle though not side dependent (main effect for muscle: *P* < 0.005; for side: *P* = 0.51). While rectus femoris was active for the longest duration (51% ± 6% of the rowing cycle), the duration of biceps femoris activity was the briefest (22% ± 5%) among muscles (*P* < 0.005 in both cases). No interaction between muscle and side was observed for both onset and duration values (*P* > 0.1).

### 3.2. Modulation of Muscle Activity in Both Legs

From the onset of activation to silencing, the amplitude of EMG envelopes changed concurrently in both legs and for all muscles tested. Significantly high correlation values (Pearson *R* > 0.78; *P* < 0.005) were observed for all muscles (Figure 4). The synchronous modulation of activity in both legs can be well observed for a representative participant in Figure 2; for all muscles, the amplitude of EMG envelopes increased and decreased roughly synchronously. Except for the soleus, tibialis anterior, and biceps femoris muscles, side differences in the peak value could, however, be observed. These differences were attenuated when normalising left and right envelopes with respect to the maximal envelope value obtained across cycles for each leg. When considering group data, the slope of regression lines calculated from the normalised envelopes did not differ significantly from unity for all muscles tested (cf. *P*-value within plots in Figure 4). These results indicate that no side differences in modulation of muscle activity were present during rowing.

## 4. Discussion

In this study we sampled EMGs bilaterally to assess which, when, and how similarly muscles in both legs are activated during indoor rowing. Key results from 20 subjects revealed all muscles assessed were both active and inactive during rowing. The onset of activation and silencing instants were different between muscles though not between body sides. As discussed below and bearing in mind that our study was devised with the aim of improving existing FES-Rowing protocols, we suggest the key muscles to target during FES Rowing are vasti and soleus during the drive phase and tibialis anterior during recovery.

### 4.1. What Are the Key, Leg Muscles Recruited During Indoor Rowing?

A note here is first necessary on why we focused analysis on the timing rather than on the amplitude of surface EMGs or on both. It should be noted that stating which muscles contribute more substantially to the generation of propulsive forces, during indoor rowing in the specific case, is not possible from the amplitude of surface EMGs. Although with normalisation of EMG envelopes we likely supressed side-differences in, e.g., electrode-skin impedance, the amplitude of surface EMGs should be not regarded as informative of the force produced by different muscles [31]. For example, differences in physiological cross-sectional area overtly preclude between muscle comparisons, as a similar, relative EMG amplitude detected for different muscles would lead to different amounts of absolute force. Although we acknowledge a substantial variation in the amplitude of EMGs during rowing (Figure 2; [11,12,13]), we used this information to exclusively assess how similarly the degree of activity varied for left-right pairs of muscles. While EMG amplitude varied by equal amounts for all muscles tested (Figure 4), qualitative side-differences were appreciated for vastus medialis and biceps femoris. Side-differences in EMG amplitude have been suggested to emerge from the asymmetric kinematic demands of sweep rowing [15]. However, we do not believe the hysteresis observed for both muscles in Figure 4 was of physiological origin. First, because differently from Janshen et al. [15], 50% of our subjects were bowside rowers. Second, notwithstanding our careful procedure for electrode positioning, local variations in activity within both muscles [24,26] may have affected our recordings. Results from Figure 4 may not be used to state muscles in both limbs contributed with equal forces to the rowing movement. These results suggest however that activity and thus force was modulated bilaterally similarly during rowing, strengthening the validity of left-right onset values reported in Figure 3. Otherwise, if left-right differences in onset values had been observed, side-differences in the variation of activity would have questioned which onset value, left or right, most appropriately represents muscle activation and silencing.

Our results indicate all leg muscles assessed were recruited at some point during the rowing cycle. Corroborating previous findings [10,13,14,15], we observed the plantar flexors and both ventral and dorsal thigh muscles were active predominantly during the drive phase (i.e., when knees are extended), from just before catch to just before finish, whereas tibialis anterior was recruited within recovery (i.e., when knees are flexed; Figure 2 and Figure 3). While it is not surprising to observe activation of quadriceps and tibialis anterior respectively during knee extension and flexion, the activation of hamstrings and plantar flexors during knee extension would at first appear to pose an inefficient mechanism. As argued by Wilson et al. [13], the conflict between hamstring and gastrocnemius recruitment during drive and their commonly conceived knee flexion action are likely apparent. Considering these muscles are biarticular, in agreement with the Lombard’s paradox, they may indeed contribute to the production of knee extension moments when the feet are fixated; it is the mechanical advantage to determine the net moment produced by two antagonist, biarticular muscles active simultaneously [32]. The knee flexion torque produced by activation of hamstrings is probably taken up by ankle plantar flexion moment generated by gastrocnemius and soleus and the knee extension moment resulting from the activation of quadriceps. It seems, therefore, the collective activation of the leg muscles indicates that all provide a substantial contribution to the effective execution of the rowing gesture, with a timed activation independent of power demands (see results for stroke rates; see also [11]). While results shown in Figure 3 appear to suggest there is not a set of key leg muscles recruited during indoor rowing, circumstances may, however, call for the identification of specific key muscles; which and when to switch a limited number of muscles on-off is of crucial importance for the successful application of FES Rowing.

### 4.2. Identification of Key Leg Muscles to Target in FES-Rowing Applications

During FES Rowing, current pulses are delivered to the leg muscles to move the knees into flexion and extension. While quadriceps are stimulated during the rowing drive phase, during recovery, stimulation is commonly delivered to the hamstring muscles [1,3,8]. Our results indicate though the hamstrings may be not the most appropriate set of muscles to target during recovery. As discussed above, the hamstrings are biarticular muscles and their moment arm is likely more efficient for hip extension than knee flexion [32,33]. Moreover, even though the trunk is secured to the backrest during FES Rowing, it is possible that a fraction of the torque elicited by stimulation of hamstrings translates into a certain degree of hip extension. This possibly explains the need of using springs or of inclining the track of the rowing machine to assist recovery [2,5]. Here, we do not propose to abandon the stimulation of hamstrings during recovery. However, given that the total number of stimulation channels is often limited to four-six, distributed between muscles in both legs, our results (Figure 3) suggest recovery could be more easily facilitated by stimulation of tibialis anterior than hamstrings. During the rowing drive phase, the decision of which muscles to stimulate is less discussable; according to previous and current results, quadriceps would seem to be the main knee extensor muscles (current results; see also [1,2,3,4,5,6]). It should be noted though that soleus was consistently activated during the drive phase, both across subjects and legs (Figure 2, Figure 3 and Figure 4). Given that the feet are secured to the foot stretcher, the ankle plantar flexion torque would be expected to extend the knee, moving the sliding seat backward. Differently from gastrocnemius, soleus spans only the ankle joint and therefore, stimulation of soleus during the drive phase should not produce any knee flexion moment. Our results would therefore seem to suggest that FES Rowing protocols should focus on the stimulation of tibialis anterior during recovery and quadriceps during drive. When the number of stimulation outputs is not a constraint, stimulation of soleus may efficiently assist knee extension during drive.

Our results substantiate not only which muscles to stimulate but also when to stimulate them during FES Rowing. Frequently, the stimulation of leg muscles is driven with an open-loop mechanism, with subjects deciding when to deliver current pulses to their muscles. A few studies have, however, proposed the implementation of closed-loop algorithms for the automatic delivery of current pulses [22], relieving subjects from the necessity of coordinating voluntary with electrically elicited muscle activation. The onset values reported in Figure 3 may provide a reference indication for the design of a real-time controller, such as the machine learning finite-state controllers proposed by [34,35]. As shown in Figure 3 and reported by others [10,15], rectus femoris is silenced after finish and this prolonged activation has been suggested to assist trunk extension and flexion from the end of drive to the beginning of recovery. Indeed, silencing of rectus femoris takes place roughly concurrently with tibialis anterior recruitment (Figure 3). Considering the trunk is presently secured, by straps, to the fixed seat backrest during FES Rowing, it seems the stimulation of plantar flexors and quadriceps and of tibialis anterior defines unequivocally the drive and recovery phases, respectively. Automatically switching between drive and recovery and vice-versa in real time must, however, account for anticipatory arm movement just prior to and after finish, an issue that remains to be elucidated in future, FES-Rowing studies.

FES rowing is a cyclical activity that can be manually controlled, typically using a thumb operated switch mounted on the rowing machine handle or boat oar [1,3,36]. Automatic control is also possible, without any manual switch, using fuzzy logic control—however, users prefer to have a switch command input as this appears less ‘robotic’ [37,38]. We are motivated to determine if EMG analysis might lead to the design of improved controllers that use muscle synergies seen in able bodied rowing. EMG patterns have been used by a number of groups to design deterministic finite state machines for FES control; multichannel EMG patterns are often obtained from able bodied subjects which provide insight into which muscles are key targets for FES and when in the cycle they are active [39,40,41]. Typically, the EMG records of able-bodied subjects are examined to suggest muscles to be used and for the occurrence of invariate events that can be related to signals derived from wearable sensors. Event detection are usually performed in real-time by repeatedly applying if-then-else type rules to these sensor derived signals and the user command inputs. We anticipate that future semi-automatic, hierarchical, finite state controllers may be synthesized based on normal muscle activation profiles suggested by EMG patterns to deliver multichannel electrical stimuli using a combination of EMG derived event detectors in combination with direct command inputs from the user. The event detection rules, which relate to the finite state transition rules, can either be hand-crafted or obtained using machine learning techniques [34,35]. For example, cyclical FES muscle activation can be reconstructed using wearable sensors and the EMG from able-bodied subjects performing the same task [42]. In some cases, FES activation of leg muscles can be directly derived using supervised learning from the EMG of upper body synergistic muscles with intact innervation after SCI. The results of the present study may also be used in self-adaptive unsupervised FES control. Previously, we have demonstrated accelerating the learning rate, i.e., the number of cycles to reach a control solution, taken by a reinforcement-learning algorithm. This was achieved by pre-training the algorithm using a biomechanical model of the FES controlled motion [43]. It may now be possible to further extend this technique by using the results presented here to direct the reinforcement-learning algorithm to solutions that are closer to the normal patterns of muscle activity, presumably leading to a more smooth gesture during FES rowing.

### 4.3. A Few Considerations on the Validity of Current Results

Although the results presented here were obtained from able-bodied elite rowers, moving freely their trunk, paraplegic subjects engaged in FES rowing are not necessarily expert rowers and have their trunk secured to the backrest. Notwithstanding these differences, we do not see any reason to discredit the validity of our results. It should be noted we use our results to propose a collective reflection on which leg muscles to stimulate in FES-Rowing applications, hoping to optimise the health benefits of FES Rowing [3,6] and to increase its feasibility and therefore, its popularity among SCI individuals. Given our common experience that motor proficiency arises with the suppression of unnecessary muscle activity (i.e., cocontraction), shaping the stimulation of leg muscles according to the excitation of leg muscles observed for elite than novice rowers is more likely to delay fatigue development and to produce smooth kinematic profiles [16,44]. Even though muscle synergies elicited during rowing seem not to depend on rowers’ experience [10], the timing and profiles of muscle excitation revealed by surface EMGs in our elite rowers are presumably associated with a most efficient rowing gesture. A remaining issue is whether the results reported in Figure 3 would change had we constrained trunk movement. Except for rectus femoris, which activation lasted up to roughly mid recovery and was possibly related to control of trunk extension-flexion as discussed above, we do not see any reason to believe the predominant activation of vasti and soleus during drive and of tibialis anterior during recovery would change should our elite rowers have performed without moving their trunk. While we value devising future experiments to address this issue, the validity of our results for FES-Rowing applications likely holds.

## Figures and Tables

**Figure 1 sensors-20-01666-f001:**
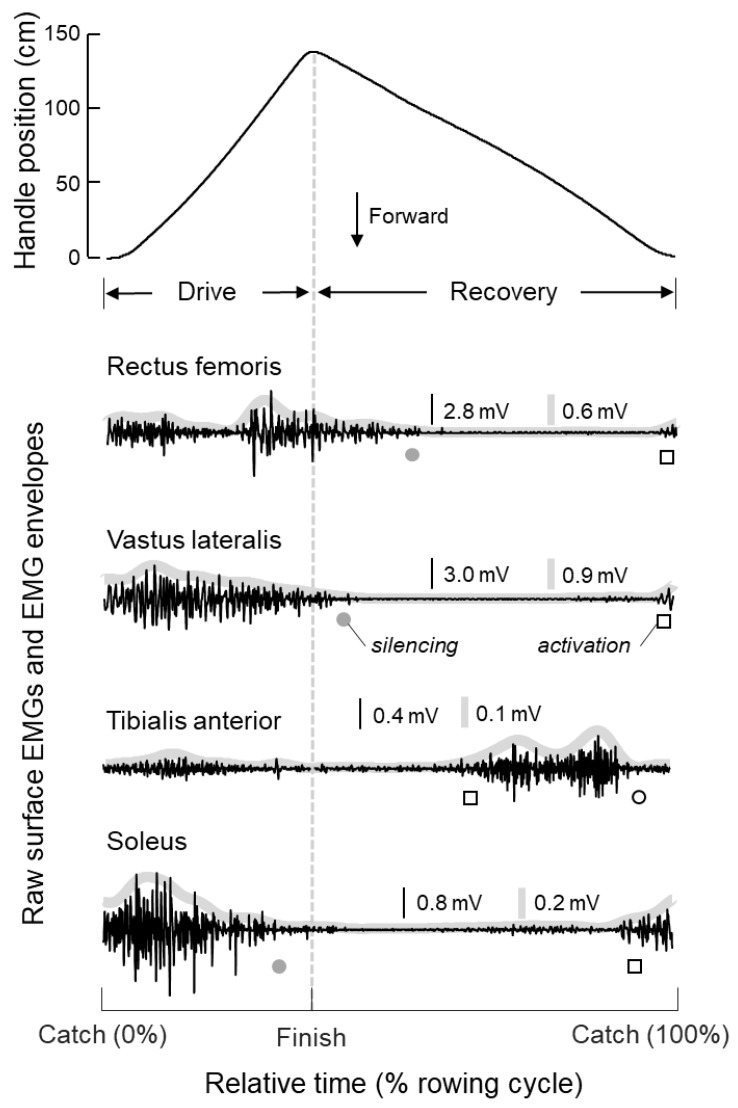
Position of the handle of the rowing machine (top) and raw surface electromyograms (EMGs) (black traces) are shown for a single subject and rowing cycle. EMGs were detected, from top to bottom, from the rectus femoris, vastus lateralis, tibialis anterior, and soleus muscles of the left body side. Grey traces superimposed on the raw EMGs denote EMG envelopes (cf. Methods). Instants corresponding to the onset of activation (squares) and to the onset of silencing computed for the current (white circle) and for the preceding rowing cycle (grey circle) are shown just below to each EMG trace. Time is represented in relative units, with respect to the rowing cycle, with the vertical dashed line indicating the finish instant.

**Figure 2 sensors-20-01666-f002:**
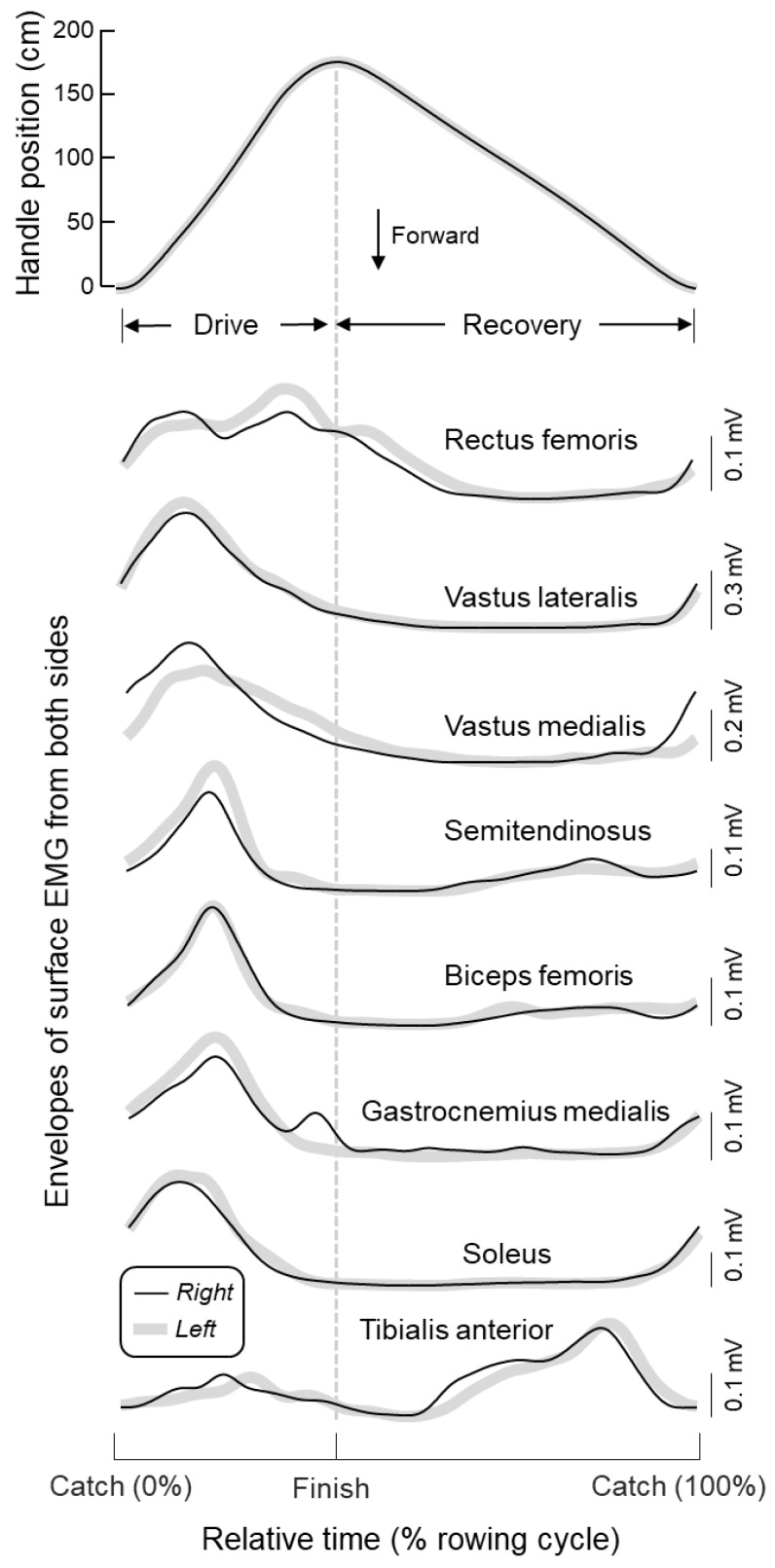
Average (thin black line) and standard deviation (thick gray line) values for the handle position across rowing cycles are shown on top for a single participant. The envelope of surface EMGs averaged across rowing cycles is shown separately for each of the eight muscles tested in the left (thin black line) and right (thick grey line) body sides. From top to bottom: Rectus femoris, vastus lateralis and medialis, semitendinosus, biceps femoris, gastrocnemius medialis, soleus and tibialis anterior. Time is represented in relative units, with respect to the rowing cycle, with the vertical dashed line indicating the finish instant. The average standard deviation for EMG envelopes was remarkably small, ranging from 19.5 to 51.5 µV across muscles and body sides.

**Figure 3 sensors-20-01666-f003:**
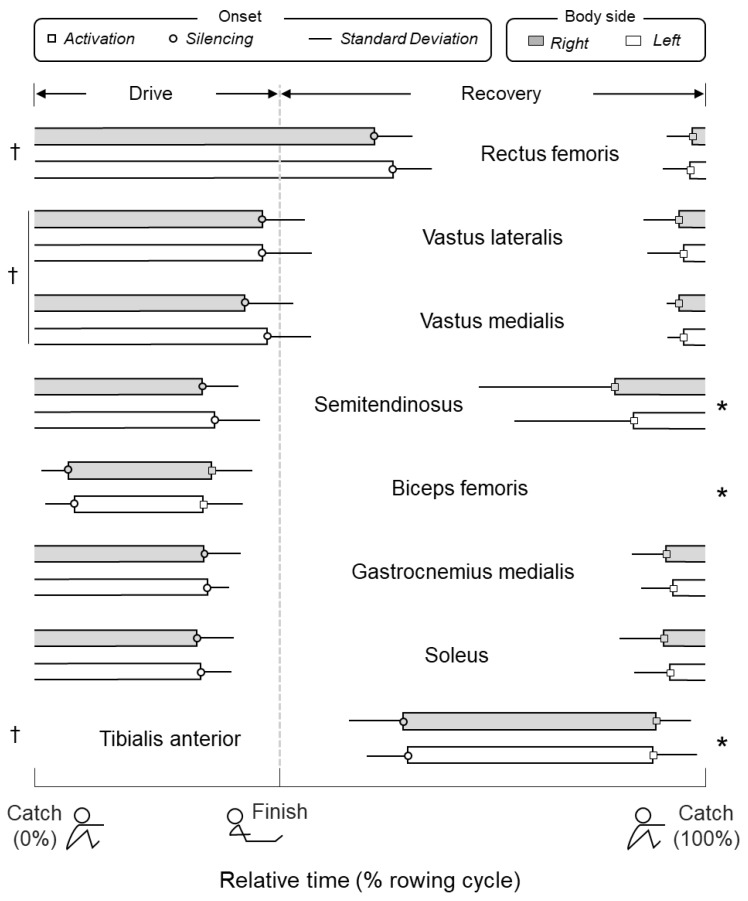
Mean values for the onset of activation (squares) and silencing (circles) are shown for the eight muscles tested in the left (white bars) and right (grey bars) sides. Horizontal lines denote the standard deviation of onset values. * and † respectively indicate the onset values of activation and silencing for the corresponding muscle were different from all other muscles (*P* < 0.05 for all cases), regardless of the body side.

**Figure 4 sensors-20-01666-f004:**
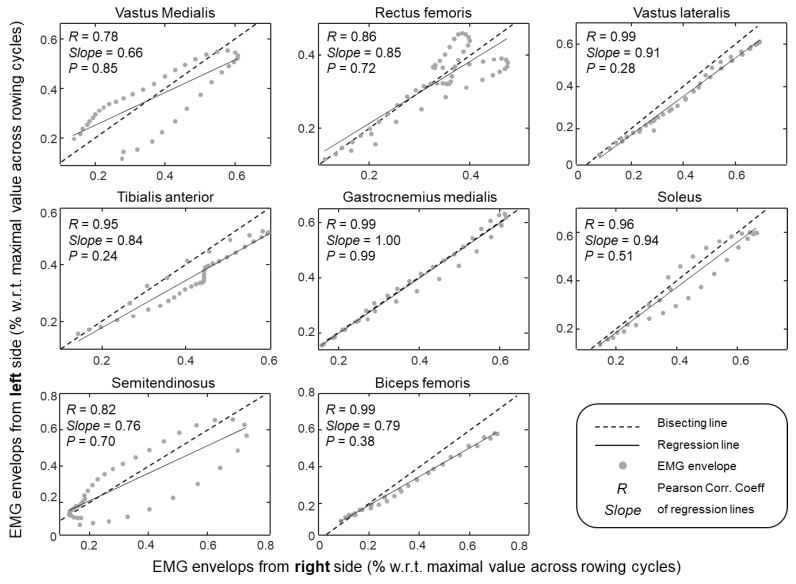
Scatter plots created from EMG envelopes (circles) obtained for periods within which muscles were active and averaged across subjects. Envelopes obtained for the left and right sides are respectively plotted along the ordinate and abscissa. Pearson correlation coefficients (*R* values) and the slope of regression (black continuous) lines are shown within each panel. Bisecting (black dashed) lines are also shown to readily indicate how much the slope of regression lines differed from unity (P-values within the plots indicate whether this difference is statistically significant).

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
