# Peer review of "Timing and Modulation of Activity in the Lower Limb Muscles During Indoor Rowing: What Are the Key Muscles to Target in FES-Rowing Protocols?"

_sensors, 2020, doi:10.3390/s20061666_

Round 1

Reviewer 1 Report

The authors quantified the onset of activation and silencing, the duration of activity of 8 lower limb muscles during indoor rowing, and how similarly these muscles were activated between lower limbs.

The study appeared to be well conducted, but I have some significant concerns about the results, and hence the contributions. The analysis was rather simple and not enough to conclude that no side differences in modulation of muscle activity were present during rowing. As the authors mentioned in Discussion, the results may change since trunk movement was constrained in FES-rowing.

Several clarifications and improvement to the text are required:

Introduction:

L52. Muscle synergy also contains time information. What’s the difference between methods proposed this paper and muscle synergies? Why not use muscle synergy to study the modulation of muscle activity?

Material and Methods:

L 85.Please provide the reference number of the ethics approval. L 90. How the three trials were combined? Did the speed and duration influence the onset and duration of muscles? L 118. How to ensure the position data was sampled synchronously with the surface EMGs? In Figure. 2, do the black and grey lines represent average EMGs from one person? Please add the deviation across rowing cycles.

Results:

Do the different circles within a subplot represent EMG envelopes of one specific subject? If so, better provide the Pearson Corr. Coeff of all subjects (mean ± std) to give a more convincing conclusion. Since the Figure 4 only showed the EMG envelopes when activated, the temporal information was ignored. More statics on the onset and duration of EMG between body sides should be provided to draw the conclusion that no side differences in modulation of muscle activity were present during rowing.

Author Response

The authors quantified the onset of activation and silencing, the duration of activity of 8 lower limb muscles during indoor rowing, and how similarly these muscles were activated between lower limbs.

The study appeared to be well conducted, but I have some significant concerns about the results, and hence the contributions. The analysis was rather simple and not enough to conclude that no side differences in modulation of muscle activity were present during rowing. As the authors mentioned in Discussion, the results may change since trunk movement was constrained in FES-rowing.

Several clarifications and improvement to the text are required:

We thank the reviewer for the comments, which we believe helped us in producing a stronger manuscript. Please find below our replies and indications of modification to the points raised.

Introduction:

L52. Muscle synergy also contains time information. What’s the difference between methods proposed this paper and muscle synergies? Why not use muscle synergy to study the modulation of muscle activity?

We thank the reviewer for raising this point. While we acknowledge the temporal profile of excitation is accounted for in the muscle synergies, we were mostly concerned with the onset of excitation and silencing. Another issue is that we feel it difficult use the synergies to rate the degree of contribution of different muscles during rowing. Even though there are criteria to define the number of synergies mostly explaining a certain motor task (rowing in our case), we feel it difficult to ascertain the most important muscle to consider within each synergy identified. For this reason we selected a more general approach and descriptive, based on the EMG amplitude itself.

Material and Methods:

L 85.Please provide the reference number of the ethics approval. L 90. How the three trials were combined? Did the speed and duration influence the onset and duration of muscles? L 118. How to ensure the position data was sampled synchronously with the surface EMGs? In Figure. 2, do the black and grey lines represent average EMGs from one person? Please add the deviation across rowing cycles.

L85: Reference number is now indicated

L90: Thank you for this observation. We mistakenly indicated subjects as a factor in our three-way statistical model. We corrected this issue indicating the factors included in the model were stroke rate, body side and muscles). As stated in the end of the first Results paragraph, there was no effect of speed on the excitation onset.

L118: Thank you for the observation. Both the biomechanical data and the EMGs signals were collected using the same system (DuePro, OT Bioelettronica, Italy). This system allows to acquire a combination of EMGs, raw signals (e.g. coming from an encoder) or conditioned, differential signals (e.g. coming from a load cells). The synchronization was achieved giving a common pulse trigger at the start and at the end of each acquisition. The common pulse trigger was wirelessly provided by the charging base of the DuePro system and allows to re-align the signals with a maximum time delay of ± one sampling period (488 µs in our case). In the revised version of the manuscript, we specified that both EMGs and biomechanical data were acquired with the same system.

Figure 2: Thank you. We explicitly indicate in the Figure caption the traces are average for a single participant. We decided though not to indicate the standard deviation in the Figure because the left-right average envelopes would be not clearly appreciated as they are now and because variability between cycles was remarkably small. In agreement with the reviewer comment, in the caption of Figure 2 we indicated the minimum and maximum average value of standard deviation for EMG envelopes across muscles and body sides.

Results:

Do the different circles within a subplot represent EMG envelopes of one specific subject? If so, better provide the Pearson Corr. Coeff of all subjects (mean ± std) to give a more convincing conclusion. Since the Figure 4 only showed the EMG envelopes when activated, the temporal information was ignored. More statics on the onset and duration of EMG between body sides should be provided to draw the conclusion that no side differences in modulation of muscle activity were present during rowing. 

No, circles in Figure 4 correspond to the mean value across subjects for each specific time point, from recruitment to silencing. We explicitly indicate this in the caption of Figure 4. In statistics we explicitly state the procedure used for the compulation of Pearson correlation “Before computing Pearson correlation, left and right envelopes were interpolated (N=100 data points), from recruitment to silencing, and normalised with respect to the maximal value across all cycles for each leg and subject to suppress the effect of spurious factors (e.g., side-differences in electrode-skin impedance; cf. [1]).”

In Figure 3 we show instants of recruitment and silencing for all muscles tested on both sides. In Figure 4 we show that within these periods of activity, envelopes for each muscle were modulated similarly between sides. We feel these results fully characterise the symmetric activation of the lower limb muscles investigated in this study.

Reviewer 2 Report

In this paper, authors have utilized surface electromyography technique to investigate the timing of activation and silencing of lower limb muscles during indoor rowing. Their goal is to recognize target muscles for application of functional electrical stimulation during rowing. They have tested twenty healthy human subjects and analyzed the data. They have also provided a good discussion to show that the result of their research is important for consideration regarding FES-rowing protocols. However, they have not conducted any FES experiments on human patients, and I believe without data from such experiments, we cannot reach to a conclusion since patients’ muscles responses to stimulation may reveal some unexpected results.

Surface electromyography is a very known technique for investigation of muscle activities. Authors have not developed any new technology or method for their experiments. Regarding journal aim and scope, I found no novelty related to sensors and sensor technologies or applications in this paper.

Author Response

In this paper, authors have utilized surface electromyography technique to investigate the timing of activation and silencing of lower limb muscles during indoor rowing. Their goal is to recognize target muscles for application of functional electrical stimulation during rowing. They have tested twenty healthy human subjects and analyzed the data. They have also provided a good discussion to show that the result of their research is important for consideration regarding FES-rowing protocols. However, they have not conducted any FES experiments on human patients, and I believe without data from such experiments, we cannot reach to a conclusion since patients’ muscles responses to stimulation may reveal some unexpected results.

We agree that FES tests are required to explore the proposed improvements. This will be the subject of future reports of our on-going FES rowing research based on the present findings. We amended discussion to emphasise the potentialities of our results.

Surface electromyography is a very known technique for investigation of muscle activities. Authors have not developed any new technology or method for their experiments. Regarding journal aim and scope, I found no novelty related to sensors and sensor technologies or applications in this paper.

The present study identifies key rowing events and proposes new muscle activation profiles for FES control. This suggests the need for new sensor and real time analysis for FES control. For example, real-time detection algorithms for the key events observed in the able-bodied rowers. Such detectors could be based on EMG patterns from muscles that retain their innervation after SCI or by using motion sensors such as IMUs. In any case, detection rules need be developed, most likely using either hand-crafted or machine learning techniques (our references 34 and 35). It is anticipated that these event detectors would then be incorporated into FES controllers such as fixed finite-state (our reference 22). The events and muscle activation profiles found in the present study could be used as the initial conditions for self-adaptive control based on machine learning (our new reference 43). We propose to investigate jump-starting reinforcement learning control (our new reference 44), where the observed muscle activation profile in able-bodied rowers can be used to direct a reinforcement-learning algorithm to solutions based on normal patterns of muscle activity.

Reviewer 3 Report

This paper looks at the timings of muscle activation during stationary rowing performed by elite rowers. The idea is to use the patterns to guide development and selection of the muscle stimulated and their timings during FES rowing. This concept has been tried with FES cycling and to my knowledge this hasn’t has not resulted in much gain at all. To some extent, success depends on what the actual goal is.

Most times FES studies lump all the different outcomes together, for example previous studies often state that their modifications/research are expected to achieve a better rowing style (or cycling), produce more power, and it will also improve health more. However, they never present a strong case on why we would expect all these outcomes to be met and improved and confuse better movement action with better health outcomes.

In this case I think you need to be clearer on what you are trying to achieve and how. In the current case the findings suggest the stimulation of the TA which is a very small muscle. How will this improve health benefits? Wouldn’t stimulating the gluteal muscles better?

The use of elite rowers will interest those who are interested in sports performance, but isn’t actually that relevant when comparing to persons with paraplegic (who are generally not elite) and ES exercise. Any sort of able bodied rower would have been suitable and perhaps less elite more relevant. Getting these rowers to train and perform on the FES rower with the trunk constraint would be very interesting; perhaps this should be considered in future studies.

Section 4.1 The material on asymmetry seems less relevant to FES rowing.

Can you make it clearer how does the muscle activity measured varies from that currently used in FES rowing. Or is it now consistent across FES rowing systems?

The English needs improving throughout. Delete unnecessary words throughout, especially at the start of sentences. Some odd word choices are also included e.g. “instalment of fatigue” or Line 273 “factual results”. Other examples: Line 210 improve “could however be appreciated”, Line 292-293 reword sentence, Line 294 “fixated”

You should include the ethics approval number and the governing body.

Specific comments:

Line 39 – what are the musculoskeletal & cardiovascular benefits of FES rowing? Please state.

Line 42 / line 102 Why are the gluteals not stimulated during FES rowing or measured here? Do elite rowers not use their gluteals?

Line 66 Are your proposing this for a system with a moving trunk … its not clear in the rest of the paper that this is the goal?

Line 75 It might help performance but please justify how this will help improve the health benefits? The two are not the same necessarily.

Line 93 How was the order of trials randomized? What method was used?

Line 97 what does “consistently low” mean exactly?

Line 177. Cadences not cadence.

Line 178. You could represent the means as stroke per minute not seconds per stroke. That might mean more to the reader.

Discussion

Line 229 & Section 4.2 298-300.

How will stimulating such small muscles contribute to health benefits? They will not help increase the VO2 much.

Line 301 Your results suggest which muscles to stimulate to mimic voluntary rowing. Do you expect this to transfer to FES considering a different fatigue rate?

308 – Is decruited a word? Decruitment as well.

Line 338 – health benefits – what is your logic? How does refinement of stimulation angle increase health benefits? Evidence for this? Aerobic response from the FES is largely dependent on the stimulation intensity (pulse amplitude & pulse width), the number of channels stimulated, and the size of those muscles. Muscle growth will depend on the forces applied.

Popularity – why would this increase popularity? Perhaps instead you wish to increase the feasibility of rowing for more people with paraplegia. That might improve health as they can now do it more easily (& hence more often?).

Line 339 Stimulation based on voluntary activation – how will this decrease fatigue. The reasons for fatigue are different than voluntary contractions and the fatigue rate in ES is much more.

Author Response

This paper looks at the timings of muscle activation during stationary rowing performed by elite rowers. The idea is to use the patterns to guide development and selection of the muscle stimulated and their timings during FES rowing. This concept has been tried with FES cycling and to my knowledge this hasn’t has not resulted in much gain at all. To some extent, success depends on what the actual goal is.

Most times FES studies lump all the different outcomes together, for example previous studies often state that their modifications/research are expected to achieve a better rowing style (or cycling), produce more power, and it will also improve health more. However, they never present a strong case on why we would expect all these outcomes to be met and improved and confuse better movement action with better health outcomes.

In this case I think you need to be clearer on what you are trying to achieve and how. In the current case the findings suggest the stimulation of the TA which is a very small muscle. How will this improve health benefits? Wouldn’t stimulating the gluteal muscles better?

We thank the reviewer for the constructive criticisms raised. We agree with the reviewer that some studies on the application of FES may be unclear when it comes to the ultimate, practical goal. We nevertheless believe there are important pieces of evidence, including longitudinal design studies, posing the benefits of combining FES with indoor rowing [2–5]. Caveats exist though, as for example the determination of the most suitable muscles to stimulate, of the optimal stimulation pattern and the often early development of muscle fatigue. In this study we use surface EMG to specifically identify the muscles most likely contributing to the rowing cycle in able-bodied rowers, hoping it would help determining the muscles to target in FES-rowing protocols.

Specifically concerning the reviewer questions, please kindly consider one of the most acknowledged issues with FES rowing is the recovery phase. In the World Rowing Championship there has been introduced indeed the “assisted return FES” category, whereby assistants pull rowers from the finish all way back to catch. Assisting return, that is ensuring subjects complete the recovery phase, means rowers must flex their knee until reaching catch. On this regard, we are unsure on whether stimulation of gluteal muscles would be better in relation to TA; of all lower limb muscles, TA is likely the sole one which dorsiflex torque over the ankle would uniquely help rowers during drive (cf. Figure 3). During drive, our results suggest the primary knee extensors (vastus muscles) are the most promising candidates for stimulation, corroborating other studies. While we suggest the stimulation of soleus may provide an added value to stroke power during the drive phase (section 4.2), we are unsure on how to motivate the testing of a proximal muscle like gluteus.

The use of elite rowers will interest those who are interested in sports performance, but isn’t actually that relevant when comparing to persons with paraplegic (who are generally not elite) and ES exercise. Any sort of able bodied rower would have been suitable and perhaps less elite more relevant. Getting these rowers to train and perform on the FES rower with the trunk constraint would be very interesting; perhaps this should be considered in future studies.

We thank the reviewer for this important point. We specifically discuss this tissue in lines 336-351.

“Although the results presented here were obtained from able-bodied elite rowers, moving freely their trunk, paraplegic subjects engaged in FES rowing are not necessarily expert rowers and have their trunk secured to the backrest. Notwithstanding these differences, we do not see any reason to discredit the validity of our results. It should be noted we use our results to propose a collective reflection on which leg muscles to stimulate in FES-Rowing applications, hoping to optimise the health benefits of FES Rowing [3,6] and to increase its popularity. Shaping the stimulation of leg muscles according to the excitation of leg muscles observed for elite than novice rowers is more likely to delay fatigue instalment and to produce smooth kinematic profiles [16,42]. Moreover, muscle synergies elicited during rowing seem not to depend on rowers’ experience [10]. A remaining issue is whether the results reported in Figure 3 would change had we constrained trunk movement. Except for rectus femoris, which activation lasted up to roughly mid recovery and was possibly related to control of trunk extension-flexion as discussed above, we do not see any reason to believe the predominant activation of vasti and soleus during drive and of tibialis anterior during recovery would change should our elite rowers have performed without moving their trunk. While we value devising future experiments to address this issue, the validity of our results for FES-Rowing applications likely holds.”

Section 4.1 The material on asymmetry seems less relevant to FES rowing.

We are not fully sure we understand the reviewer point. When assessing the timing of muscle excitation to identify which muscle are predominantly involved in rowing, we had to make sure no side differences were present. Otherwise, if left-right differences in onset values had been observed, side-differences in the variation of activity would have questioned which onset value, left or right, most appropriately represents muscle recruitment and decruitment.

Can you make it clearer how does the muscle activity measured varies from that currently used in FES rowing. Or is it now consistent across FES rowing systems?

Also here, we sadly feel we did not fully get the reviewer point. Here we measured EMGs and from them we infer on timing and degree of muscle excitation. In FES rowing system muscle excitation is induced externally, through stimulation (current/voltage) pulses applied typically transcutaneously.

The English needs improving throughout. Delete unnecessary words throughout, especially at the start of sentences. Some odd word choices are also included e.g. “instalment of fatigue” or Line 273 “factual results”. Other examples: Line 210 improve “could however be appreciated”, Line 292-293 reword sentence, Line 294 “fixated”

We thank the reviewer for this point. We double checked the manuscript to comply with the reviewer request.

You should include the ethics approval number and the governing body.

Thank you, we added specific information regarding ethics approval.

Specific comments:

Line 39 – what are the musculoskeletal & cardiovascular benefits of FES rowing? Please state.

We added specific examples in the sentence. Thank you.

Line 42 / line 102 Why are the gluteals not stimulated during FES rowing or measured here? Do elite rowers not use their gluteals?

Please, see the second paragraph of our response to the first reviewer comment.

Line 66 Are your proposing this for a system with a moving trunk … its not clear in the rest of the paper that this is the goal?

In agreement with the reviewer comment, we reworded this sentence with:

“In this study we investigate how many, when and how different leg muscles are recruited during indoor rowing, expecting our results may contribute to the definition of which muscles to target during FES-Rowing protocols”

Line 75 It might help performance but please justify how this will help improve the health benefits? The two are not the same necessarily.

We added references to this clause, both for healthy benefits and performance. Our reasoning is in agreement with the notion that:

exercise intensity during FES Rowing approaches, and possible exceed [3], the threshold recommended by the American College of Sports Medicine for reducing the risk of coronary heart disease [6]. FES Rowing may decrease the loss of bone mineral density [5].

Line 93 How was the order of trials randomized? What method was used?

We used the Matlab randperm function, whereby non-repeating integers are defined by a uniform, pseudorandom number generator.

Line 97 what does “consistently low” mean exactly?

This means subjects were asked to maintain a constant stroke power output throughout each trial. It is common practice though that rowers monitor power through the projected time taken to complete 500 m. We slightly reworded this sentence to indicate subjects “were instructed to keep consistently low the time taken to cover 500 m throughout each of the three trials.”

Line 177. Cadences not cadence.

Thank you, we corrected as suggested.

Line 178. You could represent the means as stroke per minute not seconds per stroke. That might mean more to the reader.

We thank the reviewer for this observation. Even though we agree using stroke rates may be more easily appreciated by the general population, we feel using a fractional number to represent the duration of rowing cycles would be misleading.

Discussion

Line 229 & Section 4.2 298-300.

How will stimulating such small muscles contribute to health benefits? They will not help increase the VO2 much.

We feel it difficult to judge increases in aerobic capacity. We also disagree on the view that these are small muscles, in particular when considering vasti and soleus. Based on the notion that FES Rowing exercise intensities much higher than those reported for other modalities of hybrid exercise, and the suggested and motivated involvement of TA during recovery, we therefore do not believe our results should be discredited.

Line 301 Your results suggest which muscles to stimulate to mimic voluntary rowing. Do you expect this to transfer to FES considering a different fatigue rate?

Yes, we expect our results on muscle activation during voluntary contraction in able-bodied rowers could be successfully transferred to FES-Rowing practice. Considering the ultimate goal of moving the knees into flexion and extension, as discussed in section 4.3, we believe pattern of muscle excitation observed in our elite rowers is likely to help e.g. delaying fatigue, a cardinal issue in FES Rowing [4].

308 – Is decruited a word? Decruitment as well.

Even though decruit, decruited decruitment are words often used in motor unit and muscle literature, we agree these may be not as clear as muscle silencing or deactivation. We reworded the text accordingly.

Line 338 – health benefits – what is your logic? How does refinement of stimulation angle increase health benefits? Evidence for this? Aerobic response from the FES is largely dependent on the stimulation intensity (pulse amplitude & pulse width), the number of channels stimulated, and the size of those muscles. Muscle growth will depend on the forces applied.

When it comes to aerobic capacity, it is our understanding that greater improvements arise from the involvement of a larger muscle mass under more prolonged efforts (McArdle et al 2010). As discussed in our manuscript, we expect the inclusion of TA and SOL may lead to a more active rowing cycle and help delaying fatigue. In agreement with the reviewer next concern and with our discussion, if paraplegic subjects are able to more efficiently coordinate voluntary with electrically elicited contractions, producing a more powerful stroke, we believe health benefits induced by hybrid exercises are likely to be optimised. As per the second question, we are not sure what the reviewer means by refinement of stimulation angle.

McArdle WD, Katch FI, Katch VL. Exercise Physiology: Nutrition, Energy and Human Performance. Lippincott Williams & Wilkins; 8 edizione. 2014.

Popularity – why would this increase popularity? Perhaps instead you wish to increase the feasibility of rowing for more people with paraplegia. That might improve health as they can now do it more easily (& hence more often?).

It is our hope that by making FES Rowing as close as possible to indoor and possibly on-water rowing, the number of SCI subjects adhering to this exercise paradigm will increase. We amended this statement in agreement with the reviewer concern.

Line 339 Stimulation based on voluntary activation – how will this decrease fatigue. The reasons for fatigue are different than voluntary contractions and the fatigue rate in ES is much more.

Following the reviewer concern, we reworded this paragraph to make this point clear. We believe the main difference for fatigue development is the number of muscles involved in FES and normal rowing. While in normal rowing able-bodied subjects coordinate the activaiton of several leg muscles, during FES Rowing movement is induced by a limited set of muscles. This leaves FES Rowing users with the constraint of having to rely on high intensity and frequency stimulation profiles to induce knee joint movement as similar as possible to that observed in normal rowing.

Round 2

Reviewer 1 Report

The author has made some improvements to the questions raised. However, the research methods, experiments and results in this paper still fail to provide sufficient support for its conclusion. The main concern about this paper remains unsolved. 

Author Response

The author has made some improvements to the questions raised. However, the research methods, experiments and results in this paper still fail to provide sufficient support for its conclusion. The main concern about this paper remains unsolved.

We thank the reviewer for the feedback.  We nevertheless believe to have specifically addressed all concerns raised by the reviewer, providing references or compelling arguments in support of our view when appropriate.  We would be pleased to consider further revising our manuscript should the reviewer kindly indicate and motivate the main concerns we failed to address in the previous round.